# Understanding variable importances in forests of randomized trees

**Gilles Louppe, Louis Wehenkel, Antonio Sutera and Pierre Geurts**
Dept. of EE & CS, University of Liège, Belgium
{g.louppe, l.wehenkel, a.sutera, p.geurts}@ulg.ac.be

## Abstract

Despite growing interest and practical use in various scientific areas, variable importances derived from tree-based ensemble methods are not well understood from a theoretical point of view. In this work we characterize the Mean Decrease Impurity (MDI) variable importances as measured by an ensemble of totally randomized trees in asymptotic sample and ensemble size conditions. We derive a three-level decomposition of the information jointly provided by all input variables about the output in terms of i) the MDI importance of each input variable, ii) the degree of interaction of a given input variable with the other input variables, iii) the different interaction terms of a given degree. We then show that this MDI importance of a variable is equal to zero if and only if the variable is irrelevant and that the MDI importance of a relevant variable is invariant with respect to the removal or the addition of irrelevant variables. We illustrate these properties on a simple example and discuss how they may change in the case of non-totally randomized trees such as Random Forests and Extra-Trees.

## 1 Motivation

An important task in many scientific fields is the prediction of a response variable based on a set of predictor variables. In many situations though, the aim is not only to make the most accurate predictions of the response but also to identify which predictor variables are the most important to make these predictions, e.g. in order to understand the underlying process. Because of their applicability to a wide range of problems and their capability to both build accurate models and, at the same time, to provide variable importance measures, Random Forests (Breiman, 2001) and variants such as Extra-Trees (Geurts et al., 2006) have become a major data analysis tool used with success in various scientific areas.

Despite their extensive use in applied research, only a couple of works have studied the theoretical properties and statistical mechanisms of these algorithms. Zhao (2000), Breiman (2004), Biau et al. (2008); Biau (2012), Meinshausen (2006) and Lin and Jeon (2006) investigated simplified to very realistic variants of these algorithms and proved the consistency of those variants. Little is known however regarding the variable importances computed by Random Forests like algorithms, and – as far as we know – the work of Ishwaran (2007) is indeed the only theoretical study of tree-based variable importance measures.

In this work, we aim at filling this gap and present a theoretical analysis of the Mean Decrease Impurity importance derived from ensembles of randomized trees. The rest of the paper is organized as follows: in section 2, we provide the background about ensembles of randomized trees and recall how variable importances can be derived from them; in section 3, we then derive a characterization in asymptotic conditions and show how variable importances derived from totally randomized trees offer a three-level decomposition of the information jointly contained in the input variables about the output; section 4 shows that this characterization only depends on the relevant variables and section 5

discusses theses ideas in the context of variants closer to the Random Forest algorithm; section 6 then illustrates all these ideas on an artificial problem; finally, section 7 includes our conclusions and proposes directions of future works.

## 2 Background

In this section, we first describe decision trees, as well as forests of randomized trees. Then, we describe the two major variable importances measures derived from them – including the Mean Decrease Impurity (MDI) importance that we will study in the subsequent sections.

### 2.1 Single classification and regression trees and random forests

A binary classification (resp. regression) tree (Breiman et al., 1984) is an input-output model represented by a tree structure $T$, from a random input vector $(X_1, ..., X_p)$ taking its values in $\mathcal{X}_1 \times ... \times \mathcal{X}_p = \mathcal{X}$ to a random output variable $Y \in \mathcal{Y}$. Any node $t$ in the tree represents a subset of the space $\mathcal{X}$, with the root node being $\mathcal{X}$ itself. Internal nodes $t$ are labeled with a binary test (or split) $s_t = (X_m < c)$ dividing their subset in two subsets[1] corresponding to their two children $t_L$ and $t_R$, while the terminal nodes (or leaves) are labeled with a best guess value of the output variable[2]. The predicted output $\hat{Y}$ for a new instance is the label of the leaf reached by the instance when it is propagated through the tree. A tree is built from a learning sample of size $N$ drawn from $P(X_1, ..., X_p, Y)$ using a recursive procedure which identifies at each node $t$ the split $s_t = s^*$ for which the partition of the $N_t$ node samples into $t_L$ and $t_R$ maximizes the decrease

$$\Delta i(s, t) = i(t) - p_L i(t_L) - p_R i(t_R) \tag{1}$$

of some impurity measure $i(t)$ (e.g., the Gini index, the Shannon entropy, or the variance of $Y$), and where $p_L = N_{t_L}/N_t$ and $p_R = N_{t_R}/N_t$. The construction of the tree stops , e.g., when nodes become pure in terms of $Y$ or when all variables $X_i$ are locally constant.

Single trees typically suffer from high variance, which makes them not competitive in terms of accuracy. A very efficient and simple way to address this flaw is to use them in the context of randomization-based ensemble methods. Specifically, the core principle is to introduce random perturbations into the learning procedure in order to produce several different decision trees from a single learning set and to use some aggregation technique to combine the predictions of all these trees. In Bagging (Breiman, 1996), trees are built on random bootstrap copies of the original data, hence producing different decision trees. In Random Forests (Breiman, 2001), Bagging is extended and combined with a randomization of the input variables that are used when considering candidate variables to split internal nodes $t$. In particular, instead of looking for the best split $s^*$ among all variables, the Random Forest algorithm selects, at each node, a random subset of $K$ variables and then determines the best split over these latter variables only.

### 2.2 Variable importances

In the context of ensembles of randomized trees, Breiman (2001, 2002) proposed to evaluate the importance of a variable $X_m$ for predicting $Y$ by adding up the weighted impurity decreases $p(t)\Delta i(s_t, t)$ for all nodes $t$ where $X_m$ is used, averaged over all $N_T$ trees in the forest:

$$Imp(X_m) = \frac{1}{N_T} \sum_T \sum_{t \in T : v(s_t) = X_m} p(t)\Delta i(s_t, t) \tag{2}$$

and where $p(t)$ is the proportion $N_t/N$ of samples reaching $t$ and $v(s_t)$ is the variable used in split $s_t$. When using the Gini index as impurity function, this measure is known as the *Gini importance* or *Mean Decrease Gini*. However, since it can be defined for any impurity measure $i(t)$, we will refer to Equation 2 as the *Mean Decrease Impurity* importance (MDI), no matter the impurity measure $i(t)$. We will characterize and derive results for this measure in the rest of this text.

In addition to MDI, Breiman (2001, 2002) also proposed to evaluate the importance of a variable $X_m$ by measuring the *Mean Decrease Accuracy* (MDA) of the forest when the values of $X_m$ are randomly permuted in the out-of-bag samples. For that reason, this latter measure is also known as the *permutation importance*.

Thanks to popular machine learning softwares (Breiman, 2002; Liaw and Wiener, 2002; Pedregosa et al., 2011), both of these variable importance measures have shown their practical utility in an increasing number of experimental studies. Little is known however regarding their inner workings. Strobl et al. (2007) compare both MDI and MDA and show experimentally that the former is biased towards some predictor variables. As explained by White and Liu (1994) in case of single decision trees, this bias stems from an unfair advantage given by the usual impurity functions $i(t)$ towards predictors with a large number of values. Strobl et al. (2008) later showed that MDA is biased as well, and that it overestimates the importance of correlated variables – although this effect was not confirmed in a later experimental study by Genuer et al. (2010). From a theoretical point of view, Ishwaran (2007) provides a detailed theoretical development of a simplified version of MDA, giving key insights for the understanding of the actual MDA.

## 3 Variable importances derived from totally randomized tree ensembles

Let us now consider the MDI importance as defined by Equation 2, and let us assume a set $V = \{X_1, ..., X_p\}$ of *categorical* input variables and a *categorical* output $Y$. For the sake of simplicity we will use the Shannon entropy as impurity measure, and focus on totally randomized trees; later on we will discuss other impurity measures and tree construction algorithms.

Given a training sample $\mathcal{L}$ of $N$ joint observations of $X_1, ..., X_p, Y$ independently drawn from the joint distribution $P(X_1, ..., X_p, Y)$, let us assume that we infer from it an infinitely large ensemble of *totally randomized and fully developed trees*. In this setting, a totally randomized and fully developed tree is defined as a decision tree in which each node $t$ is partitioned using a variable $X_i$ picked uniformly at random among those not yet used at the parent nodes of $t$, and where each $t$ is split into $|\mathcal{X}_i|$ sub-trees, i.e., one for each possible value of $\mathcal{X}_i$, and where the recursive construction process halts only when all $p$ variables have been used along the current branch. Hence, in such a tree, leaves are all at the same depth $p$, and the set of leaves of a fully developed tree is in bijection with the set $\mathcal{X}$ of all possible joint configurations of the $p$ input variables. For example, if the input variables are all binary, the resulting tree will have exactly $2^p$ leaves.

**Theorem 1.** *The MDI importance of $X_m \in V$ for $Y$ as computed with an infinite ensemble of fully developed totally randomized trees and an infinitely large training sample is:*

$$Imp(X_m) = \sum_{k=0}^{p-1} \frac{1}{C_p^k} \frac{1}{p-k} \sum_{B \in \mathcal{P}_k(V^{-m})} I(X_m; Y|B), \qquad (3)$$

where $V^{-m}$ denotes the subset $V \setminus \{X_m\}$, $\mathcal{P}_k(V^{-m})$ is the set of subsets of $V^{-m}$ of cardinality $k$, and $I(X_m; Y|B)$ is the conditional mutual information of $X_m$ and $Y$ given the variables in $B$.

*Proof.* See Appendix B. □

**Theorem 2.** *For any ensemble of fully developed trees in asymptotic learning sample size conditions (e.g., in the same conditions as those of Theorem 1), we have that*

$$\sum_{m=1}^{p} Imp(X_m) = I(X_1, \ldots, X_p; Y). \qquad (4)$$

*Proof.* See Appendix C. □

Together, theorems 1 and 2 show that variable importances derived from totally randomized trees in asymptotic conditions provide a three-level decomposition of the information $I(X_1, \ldots, X_p; Y)$ contained in the set of input variables about the output variable. The first level is a decomposition among input variables (see Equation 4 of Theorem 2), the second level is a decomposition along the

degrees $k$ of interaction terms of a variable with the other ones (see the outer sum in Equation 3 of Theorem 1), and the third level is a decomposition along the combinations $B$ of interaction terms of fixed size $k$ of possible interacting variables (see the inner sum in Equation 3).

We observe that the decomposition includes, for each variable, each and every interaction term of each and every degree weighted in a fashion resulting only from the combinatorics of possible interaction terms. In particular, since all $I(X_m; Y|B)$ terms are at most equal to $H(Y)$, the prior entropy of $Y$, the $p$ terms of the outer sum of Equation 3 are each upper bounded by

$$\frac{1}{C_p^k} \frac{1}{p-k} \sum_{B \in \mathcal{P}_k(V^{-m})} H(Y) = \frac{1}{C_p^k} \frac{1}{p-k} C_{p-1}^k H(Y) = \frac{1}{p} H(Y).$$

As such, the second level decomposition resulting from totally randomized trees makes the $p$ sub-importance terms $\frac{1}{C_p^k} \frac{1}{p-k} \sum_{B \in \mathcal{P}_k(V^{-m})} I(X_m; Y|B)$ to equally contribute (at most) to the total importance, even though they each include a combinatorially different number of terms.

## 4 Importances of relevant and irrelevant variables

Following Kohavi and John (1997), let us define as *relevant to Y with respect to V* a variable $X_m$ for which there exists at least one subset $B \subseteq V$ (possibly empty) such that $I(X_m; Y|B) > 0$.[3] Thus we define as *irrelevant to Y with respect to V* a variable $X_i$ for which, for all $B \subseteq V$, $I(X_i; Y|B) = 0$. Remark that if $X_i$ is irrelevant to $Y$ with respect to $V$, then by definition it is also irrelevant to $Y$ with respect to any subset of $V$.

**Theorem 3.** *$X_i \in V$ is irrelevant to $Y$ with respect to $V$ if and only if its infinite sample size importance as computed with an infinite ensemble of fully developed totally randomized trees built on $V$ for $Y$ is 0.*

*Proof.* See Appendix D. □

**Lemma 4.** *Let $X_i \notin V$ be an irrelevant variable for $Y$ with respect to $V$. The infinite sample size importance of $X_m \in V$ as computed with an infinite ensemble of fully developed totally randomized trees built on $V$ for $Y$ is the same as the importance derived when using $V \cup \{X_i\}$ to build the ensemble of trees for $Y$.*

*Proof.* See Appendix E. □

**Theorem 5.** *Let $V_R \subseteq V$ be the subset of all variables in $V$ that are relevant to $Y$ with respect to $V$. The infinite sample size importance of any variable $X_m \in V_R$ as computed with an infinite ensemble of fully developed totally randomized trees built on $V_R$ for $Y$ is the same as its importance computed in the same conditions by using all variables in $V$. That is:*

$$
\begin{aligned}
Imp(X_m) &= \sum_{k=0}^{p-1} \frac{1}{C_p^k} \frac{1}{p-k} \sum_{B \in \mathcal{P}_k(V^{-m})} I(X_m; Y|B) \\
&= \sum_{l=0}^{r-1} \frac{1}{C_r^l} \frac{1}{r-l} \sum_{B \in \mathcal{P}_l(V_R^{-m})} I(X_m; Y|B)
\end{aligned}
\tag{5}
$$

*where $r$ is the number of relevant variables in $V_R$.*

*Proof.* See Appendix F. □

Theorems 3 and 5 show that the importances computed with an ensemble of totally randomized trees depends only on the relevant variables. Irrelevant variables have a zero importance and do not affect the importance of relevant variables. Practically, we believe that such properties are desirable conditions for a sound criterion assessing the importance of a variable. Indeed, noise should not be credited of any importance and should not make any other variable more (or less) important.

# 5 Random Forest variants

In this section, we consider and discuss variable importances as computed with other types of ensembles of randomized trees. We first show how our results extend to any other impurity measure, and then analyze importances computed by depth-pruned ensemble of randomized trees and those computed by randomized trees built on random subspaces of fixed size. Finally, we discuss the case of non-totally randomized trees.

## 5.1 Generalization to other impurity measures

Although our characterization in sections 3 and 4 uses Shannon entropy as the impurity measure, we show in Appendix I that theorems 1, 3 and 5 hold for other impurity measures, simply substituting conditional mutual information for conditional impurity reduction in the different formulas and in the definition of irrelevant variables. In particular, our results thus hold for the Gini index in classification and can be extended to regression problems using variance as the impurity measure.

## 5.2 Pruning and random subspaces

In sections 3 and 4, we considered totally randomized trees that were fully developed, i.e. until all $p$ variables were used within each branch. When totally randomized trees are developed only up to some smaller depth $q \leq p$, we show in Proposition 6 that the variable importances as computed by these trees is limited to the $q$ first terms of Equation 3. We then show in Proposition 7 that these latter importances are actually the same as when each tree of the ensemble is fully developed over a random subspace (Ho, 1998) of $q$ variables drawn prior to its construction.

**Proposition 6.** *The importance of $X_m \in V$ for $Y$ as computed with an infinite ensemble of pruned totally randomized trees built up to depth $q \leq p$ and an infinitely large training sample is:*

$$Imp(X_m) = \sum_{k=0}^{q-1} \frac{1}{C_p^k} \frac{1}{p-k} \sum_{B \in \mathcal{P}_k(V^{-m})} I(X_m; Y|B) \tag{6}$$

*Proof.* See Appendix G. □

**Proposition 7.** *The importance of $X_m \in V$ for $Y$ as computed with an infinite ensemble of pruned totally randomized trees built up to depth $q \leq p$ and an infinitely large training sample is identical to the importance as computed for $Y$ with an infinite ensemble of fully developed totally randomized trees built on random subspaces of $q$ variables drawn from $V$.*

*Proof.* See Appendix H. □

As long as $q \geq r$ (where $r$ denotes the number of relevant variables in $V$), it can easily be shown that all relevant variables will still obtain a strictly positive importance, which will however differ in general from the importances computed by fully grown totally randomized trees built over all variables. Also, each irrelevant variable of course keeps an importance equal to zero, which means that, in asymptotic conditions, these pruning and random subspace methods would still allow us identify the relevant variables, as long as we have a good upper bound $q$ on $r$.

## 5.3 Non-totally randomized trees

In our analysis in the previous sections, trees are built totally at random and hence do not directly relate to those built in Random Forests (Breiman, 2001) or in Extra-Trees (Geurts et al., 2006). To better understand the importances as computed by those algorithms, let us consider a close variant of totally randomized trees: at each node $t$, let us instead draw uniformly at random $1 \leq K \leq p$ variables and let us choose the one that maximizes $\Delta i(t)$. Notice that, for $K = 1$, this procedure amounts to building ensembles of totally randomized trees as defined before, while, for $K = p$, it amounts to building classical single trees in a deterministic way.

First, the importance of $X_m \in V$ as computed with an infinite ensemble of such randomized trees is not the same as Equation 3. For $K > 1$, masking effects indeed appear: at $t$, some variables are

never selected because there always is some other variable for which $\Delta i(t)$ is larger. Such effects tend to pull the best variables at the top of the trees and to push the others at the leaves. As a result, the importance of a variable no longer decomposes into a sum including all $I(X_m; Y|B)$ terms. The importance of the best variables decomposes into a sum of their mutual information alone or conditioned only with the best others – but not conditioned with all variables since they no longer ever appear at the bottom of trees. By contrast, the importance of the least promising variables now decomposes into a sum of their mutual information conditioned only with all variables – but not alone or conditioned with a couple of others since they no longer ever appear at the top of trees. In other words, because of the guided structure of the trees, the importance of $X_m$ now takes into account only some of the conditioning sets $B$, which may over- or underestimate its overall relevance.

To make things clearer, let us consider a simple example. Let $X_1$ perfectly explains $Y$ and let $X_2$ be a slightly noisy copy of $X_1$ (i.e., $I(X_1; Y) \approx I(X_2; Y)$, $I(X_1; Y|X_2) = \epsilon$ and $I(X_2; Y|X_1) = 0$). Using totally randomized trees, the importances of $X_1$ and $X_2$ are nearly equal – the importance of $X_1$ being slightly higher than the importance of $X_2$:

$$Imp(X_1) = \frac{1}{2}I(X_1; Y) + \frac{1}{2}I(X_1; Y|X_2) = \frac{1}{2}I(X_1; Y) + \frac{\epsilon}{2}$$
$$Imp(X_2) = \frac{1}{2}I(X_2; Y) + \frac{1}{2}I(X_2; Y|X_1) = \frac{1}{2}I(X_2; Y) + 0$$

In non-totally randomized trees, for $K = 2$, $X_1$ is always selected at the root node and $X_2$ is always used in its children. Also, since $X_1$ perfectly explains $Y$, all its children are pure and the reduction of entropy when splitting on $X_2$ is null. As a result, $Imp_{K=2}(X_1) = I(X_1; Y)$ and $Imp_{K=2}(X_2) = I(X_2; Y|X_1) = 0$. Masking effects are here clearly visible: the true importance of $X_2$ is masked by $X_1$ as if $X_2$ were irrelevant, while it is only a bit less informative than $X_1$.

As a direct consequence of the example above, for $K > 1$, it is no longer true that a variable is irrelevant if and only if its importance is zero. In the same way, it can also be shown that the importances become dependent on the number of irrelevant variables. Let us indeed consider the following counter-example: let us add in the previous example an irrelevant variable $X_i$ with respect to $\{X_1, X_2\}$ and let us keep $K = 2$. The probability of selecting $X_2$ at the root node now becomes positive, which means that $Imp_{K=2}(X_2)$ now includes $I(X_2; Y) > 0$ and is therefore strictly larger than the importance computed before. For $K$ fixed, adding irrelevant variables dampens masking effects, which thereby makes importances indirectly dependent on the number of irrelevant variables.

In conclusion, the importances as computed with totally randomized trees exhibit properties that do not possess, by extension, neither random forests nor extra-trees. With totally randomized trees, the importance of $X_m$ only depends on the relevant variables and is 0 if and only if $X_m$ is irrelevant. As we have shown, it may no longer be the case for $K > 1$. Asymptotically, the use of totally randomized trees for assessing the importance of a variable may therefore be more appropriate. In a finite setting (i.e., a limited number of samples and a limited number of trees), guiding the choice of the splitting variables remains however a sound strategy. In such a case, $I(X_m; Y|B)$ cannot be measured neither for all $X_m$ nor for all $B$. It is therefore pragmatic to promote those that look the most promising – even if the resulting importances may be biased.

## 6    Illustration on a digit recognition problem

In this section, we consider the digit recognition problem of (Breiman et al., 1984) for illustrating variable importances as computed with totally randomized trees. We verify that they match with our theoretical developments and that they decompose as foretold. We also compare these importances with those computed by an ensemble of non-totally randomized trees, as discussed in section 5.3, for increasing values of $K$.

Let us consider a seven-segment indicator displaying numerals using horizontal and vertical lights in on-off combinations, as illustrated in Figure 1. Let $Y$ be a random variable taking its value in $\{0, 1, ..., 9\}$ with equal probability and let $X_1, ..., X_7$ be binary variables whose values are each determined univocally given the corresponding value of $Y$ in Table 1.

Since Table 1 perfectly defines the data distribution, and given the small dimensionality of the problem, it is practicable to directly apply Equation 3 to compute variable importances. To verify our

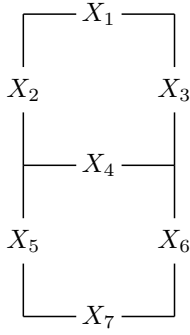

**Figure 1:** 7-segment display

| $y$ | $x_1$ | $x_2$ | $x_3$ | $x_4$ | $x_5$ | $x_6$ | $x_7$ |
|---|---|---|---|---|---|---|---|
| 0 | 1 | 1 | 1 | 0 | 1 | 1 | 1 |
| 1 | 0 | 0 | 1 | 0 | 0 | 1 | 0 |
| 2 | 1 | 0 | 1 | 1 | 1 | 0 | 1 |
| 3 | 1 | 0 | 1 | 1 | 0 | 1 | 1 |
| 4 | 0 | 1 | 1 | 1 | 0 | 1 | 0 |
| 5 | 1 | 1 | 0 | 1 | 0 | 1 | 1 |
| 6 | 1 | 1 | 0 | 1 | 1 | 1 | 1 |
| 7 | 1 | 0 | 1 | 0 | 0 | 1 | 0 |
| 8 | 1 | 1 | 1 | 1 | 1 | 1 | 1 |
| 9 | 1 | 1 | 1 | 1 | 0 | 1 | 1 |

**Table 1:** Values of $Y, X_1, ..., X_7$

|  | Eqn. 3 | $K=1$ | $K=2$ | $K=3$ | $K=4$ | $K=5$ | $K=6$ | $K=7$ |
|---|---|---|---|---|---|---|---|---|
| $X_1$ | 0.412 | 0.414 | 0.362 | 0.327 | 0.309 | 0.304 | 0.305 | 0.306 |
| $X_2$ | 0.581 | 0.583 | 0.663 | 0.715 | 0.757 | 0.787 | 0.801 | 0.799 |
| $X_3$ | 0.531 | 0.532 | 0.512 | 0.496 | 0.489 | 0.483 | 0.475 | 0.475 |
| $X_4$ | 0.542 | 0.543 | 0.525 | 0.484 | 0.445 | 0.414 | 0.409 | 0.412 |
| $X_5$ | 0.656 | 0.658 | 0.731 | 0.778 | 0.810 | 0.827 | 0.831 | 0.835 |
| $X_6$ | 0.225 | 0.221 | 0.140 | 0.126 | 0.122 | 0.122 | 0.121 | 0.120 |
| $X_7$ | 0.372 | 0.368 | 0.385 | 0.392 | 0.387 | 0.382 | 0.375 | 0.372 |
| $\sum$ | 3.321 | 3.321 | 3.321 | 3.321 | 3.321 | 3.321 | 3.321 | 3.321 |

**Table 2:** Variable importances as computed with an ensemble of randomized trees, for increasing values of $K$. Importances at $K = 1$ follow their theoretical values, as predicted by Equation 3 in Theorem 1. However, as $K$ increases, importances diverge due to masking effects. In accordance with Theorem 2, their sum is also always equal to $I(X_1, \ldots, X_7; Y) = H(Y) = \log_2(10) = 3.321$ since inputs allow to perfectly predict the output.

theoretical developments, we then compare in Table 2 variable importances as computed by Equation 3 and those yielded by an ensemble of 10000 totally randomized trees ($K = 1$). Note that given the known structure of the problem, building trees on a sample of finite size that perfectly follows the data distribution amounts to building them on a sample of infinite size. At best, trees can thus be built on a 10-sample dataset, containing exactly one sample for each of the equiprobable outcomes of $Y$. As the table illustrates, the importances yielded by totally randomized trees match those computed by Equation 3, which confirms Theorem 1. Small differences stem from the fact that a finite number of trees were built in our simulations, but those discrepancies should disappear as the size of the ensemble grows towards infinity. It also shows that importances indeed add up to $I(X_1, ...X_7; Y)$, which confirms Theorem 2. Regarding the actual importances, they indicate that $X_5$ is stronger than all others, followed – in that order – by $X_2$, $X_4$ and $X_3$ which also show large importances. $X_1$, $X_7$ and $X_6$ appear to be the less informative. The table also reports importances for increasing values of $K$. As discussed before, we see that a large value of $K$ yields importances that can be either overestimated (e.g., at $K = 7$, the importances of $X_2$ and $X_5$ are larger than at $K = 1$) or underestimated due to masking effects (e.g., at $K = 7$, the importances of $X_1$, $X_3$, $X_4$ and $X_6$ are smaller than at $K = 1$, as if they were less important). It can also be observed that masking effects may even induce changes in the variable rankings (e.g., compare the rankings at $K = 1$ and at $K = 7$), which thus confirms that importances are differently affected.

To better understand why a variable is important, it is also insightful to look at its decomposition into its $p$ sub-importances terms, as shown in Figure 2. Each row in the plots of the figure corresponds to one the $p = 7$ variables and each column to a size $k$ of conditioning sets. As such, the value at row $m$ and column $k$ corresponds the importance of $X_m$ when conditioned with $k$ other variables (e.g., to the term $\frac{1}{C_p^k} \frac{1}{p-k} \sum_{B \in \mathcal{P}_k(V^{-m})} I(X_m; Y|B)$ in Equation 3 in the case of totally randomized trees). In the left plot, for $K = 1$, the figure first illustrates how importances yielded by totally randomized trees decomposes along the degrees $k$ of interactions terms. We can observe that they each equally contribute (at most) the total importance of a variable. The plot also illustrates why $X_5$ is important: it is informative either alone or conditioned with any combination of the other variables (all of its terms are significantly larger than 0). By contrast, it also clearly shows why

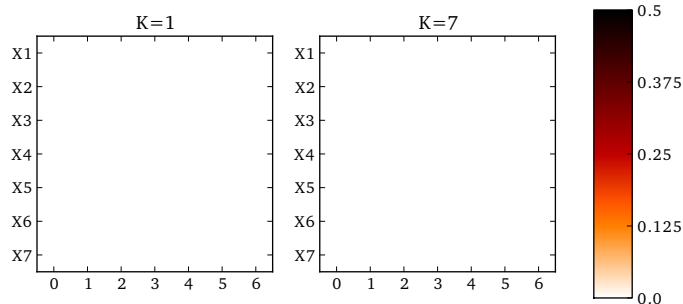

**Figure 2:** Decomposition of variable importances along the degrees $k$ of interactions of one variable with the other ones. At $K = 1$, all $I(X_m; Y|B)$ are accounted for in the total importance, while at $K = 7$ only some of them are taken into account due to masking effects.

$X_6$ is not important: neither alone nor combined with others $X_6$ seems to be very informative (all of its terms are close to 0). More interestingly, this figure also highlights redundancies: $X_7$ is informative alone or conditioned with a couple of others (the first terms are significantly larger than 0), but becomes uninformative when conditioned with many others (the last terms are closer to 0). The right plot, for $K = 7$, illustrates the decomposition of importances when variables are chosen in a deterministic way. The first thing to notice is masking effects. Some of the $I(X_m; Y|B)$ terms are indeed clearly never encountered and their contribution is therefore reduced to 0 in the total importance. For instance, for $k = 0$, the sub-importances of $X_2$ and $X_5$ are positive, while all others are null, which means that only those two variables are ever selected at the root node, hence masking the others. As a consequence, this also means that the importances of the remaining variables is biased and that it actually only accounts of their relevance when conditioned to $X_2$ or $X_5$, but not of their relevance in other contexts. At $k = 0$, masking effects also amplify the contribution of $I(X_2; Y)$ (resp. $I(X_5; Y)$) since $X_2$ (resp. $X_5$) appears more frequently at the root node than in totally randomized trees. In addition, because nodes become pure before reaching depth $p$, conditioning sets of size $k \geq 4$ are never actually encountered, which means that we can no longer know whether variables are still informative when conditioned to many others. All in all, this figure thus indeed confirms that importances as computed with non-totally randomized trees take into account only some of the conditioning sets $B$, hence biasing the measured importances.

## 7 Conclusions

In this work, we made a first step towards understanding variable importances as computed with a forest of randomized trees. In particular, we derived a theoretical characterization of the Mean Decrease Impurity importances as computed by totally randomized trees in asymptotic conditions. We showed that they offer a three-level decomposition of the information jointly provided by all input variables about the output (Section 3). We then demonstrated (Section 4) that MDI importances as computed by totally randomized trees exhibit desirable properties for assessing the relevance of a variable: it is equal to zero if and only if the variable is irrelevant and it depends only on the relevant variables. We discussed the case of Random Forests and Extra-Trees (Section 5) and finally illustrated our developments on an artificial but insightful example (Section 6).

There remain several limitations to our framework that we would like address in the future. First, our results should be adapted to binary splits as used within an actual Random Forest-like algorithm. In this setting, any node $t$ is split in only two subsets, which means that any variable may then appear one or several times within a branch, and thus should make variable importances now dependent on the cardinalities of the input variables. In the same direction, our framework should also be extended to the case of continuous variables. Finally, results presented in this work are valid in an asymptotic setting only. An important direction of future work includes the characterization of the distribution of variable importances in a finite setting.

**Acknowledgements.** Gilles Louppe is a research fellow of the FNRS (Belgium) and acknowledges its financial support. This work is supported by PASCAL2 and the IUAP DYSCO, initiated by the Belgian State, Science Policy Office.

## Footnotes

[1]More generally, splits are defined by a (not necessarily binary) partition of the range $\mathcal{X}_m$ of possible values of a single variable $X_m$.

[2]e.g. determined as the majority class $j(t)$ (resp., the average value $\bar{y}(t)$) within the subset of the leaf $t$.

[3]Among the relevant variables, we have the *marginally* relevant ones, for which $I(X_m; Y) > 0$, the *strongly* relevant ones, for which $I(X_m; Y|V^{-m}) > 0$, and the *weakly* relevant variables, which are relevant but not strongly relevant.

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
