[Supplementary Material · louppe13-suppl.pdf]

# Understanding variable importances in forests of randomized trees
## *Supplementary materials*

**Gilles Louppe, Louis Wehenkel, Antonio Sutera and Pierre Geurts**
Dept. of EE & CS, University of Liège, Belgium
{g.louppe, l.wehenkel, a.sutera, p.geurts}@ulg.ac.be

## A   Notations, and definitions of entropies and mutual information

To be self-contained, we first recall several definitions from information theory (see Cover and Thomas (2012), for further properties).

We suppose that we are given a probability space $(\Omega, \mathcal{E}, \mathbb{P})$ and consider random variables defined on it taking a finite number of possible values. We use upper case letters to denote such random variables (e.g. $X, Y, Z, W \ldots$) and calligraphic letters (e.g. $\mathcal{X}, \mathcal{Y}, \mathcal{Z}, \mathcal{W} \ldots$) to denote their image sets (of finite cardinality), and lower case letters (e.g. $x, y, z, w \ldots$) to denote one of their possible values. For a (finite) set of (finite) random variables $X = \{X_1, \ldots, X_i\}$, we denote by $P_X(x) = P_X(x_1, \ldots, x_i)$ the probability $\mathbb{P}(\{\omega \in \Omega \mid \forall \ell : 1, \ldots, i : X_\ell(\omega) = x_\ell\})$, and by $\mathcal{X} = \mathcal{X}_1 \times \cdots \times \mathcal{X}_i$ the set of joint configurations of these random variables. Given two sets of random variables, $X = \{X_1, \ldots, X_i\}$ and $Y = \{Y_1, \ldots, Y_j\}$, we denote by $P_{X|Y}(x \mid y) = P_{X,Y}(x, y)/P_Y(y)$ the conditional density of $X$ with respect to $Y$.[1]

With these notations, the joint (Shannon) entropy of a set of random variables $X = \{X_1, \ldots, X_i\}$ is thus defined by

$$H(X) = -\sum_{x \in \mathcal{X}} P_X(x) \log_2 P_X(x),$$

while the mean conditional entropy of a set of random variables $X = \{X_1, \ldots, X_i\}$, given the values of another set of random variables $Y = \{Y_1, \ldots, Y_j\}$ is defined by

$$H(X \mid Y) = -\sum_{x \in \mathcal{X}} \sum_{y \in \mathcal{Y}} P_{X,Y}(x, y) \log_2 P_{X|Y}(x \mid y).$$

The mutual information among the set of random variables $X = \{X_1, \ldots, X_i\}$ and the set of random variables $Y = \{Y_1, \ldots, Y_j\}$ is defined by

$$
\begin{aligned}
I(X; Y) &= -\sum_{x \in \mathcal{X}} \sum_{y \in \mathcal{Y}} P_{X,Y}(x, y) \log_2 \frac{P_X(x) P_Y(y)}{P_{X,Y}(x, y)} \\
&= H(X) - H(X \mid Y) \\
&= H(Y) - H(Y \mid X).
\end{aligned}
$$

The mean conditional mutual information among the set of random variables $X = \{X_1, \ldots, X_k\}$ and the set of random variables $Y = \{Y_1, \ldots, Y_j\}$, given the values of a third set of random variables $Z = \{Z_1, \ldots, Z_i\}$, is defined by

$$
\begin{aligned}
I(X; Y \mid Z) &= H(X \mid Z) - H(X \mid Y, Z) \\
&= H(Y \mid Z) - H(Y \mid X, Z)
\end{aligned}
$$

$$= -\sum_{x \in \mathcal{X}} \sum_{y \in \mathcal{Y}} \sum_{z \in \mathcal{Z}} P_{X,Y,Z}(x,y,z) \log_2 \frac{P_{X|Z}(x \mid z) P_{Y|Z}(y \mid z)}{P_{X,Y|Z}(x,y \mid z)}.$$

We also recall the chaining rule

$$I(X, Z; Y \mid W) = I(X; Y \mid W) + I(Z; Y \mid W, X),$$

and the symmetry of the (conditional) mutual information among sets of random variables

$$I(X; Y \mid Z) = I(Y; X \mid Z).$$

## B  Proof of Theorem 1

*Proof.* Let us define $p(t)$ as the proportion $N_t/N$ of samples from $\mathcal{L}$ reaching node $t$ and $p(j|t)$ as the proportion $N_{jt}/N_t$ of samples of class $j$ in $t$. By expanding Equation 1 into Equation 2 and using the entropy $H(Y|t) = -\sum_j p(j|t) \log_2(p(j|t))$ as impurity measure $i(t)$, Equation 2 can be rewritten in terms of mutual information:

$$Imp(X_m) = \frac{1}{N_T} \sum_T \sum_{t \in T : v(s_t) = X_m} p(t) I(Y; X_m|t)$$

As the size $N$ of the training sample grows to infinity, $p(t)$ becomes the (exact) probability (according to $P(X_1, \ldots, X_p, Y)$) that an object reaches node $t$, i.e., a probability $P(B(t) = b(t))$ where $B(t) = (X_{i_1}, ..., X_{i_k})$ is the subset of $k$ variables tested in the branch from the root node to the parent of $t$ and $b(t)$ is the vector of values of these variables. As the the number $N_T$ of totally randomized trees also grows to infinity, the importance of a variable $X_m$ can then be written:

$$Imp(X_m) = \sum_{B \subseteq V^{-m}} \sum_{b \in \mathcal{X}_{i_1} \times ... \times \mathcal{X}_{i_k}} \alpha(B, b, X_m, p) P(B = b) I(Y; X_m|B = b),$$

where $b$ is a set of values for the variables in $B$ and $\alpha(B, b, X_m, p)$ is the probability that a node $t$ (at depth $k$) in a totally randomized tree tests the variable $X_m$ and is such that $B(t) = B$ and $b(t) = b$.

Let us compute $\alpha(B, b, X_m, p)$. First, let us consider the probability that a node $t$ tests the variable $X_m$ and is such that the branch leading to $t$ follows a path defined, in that particular order, by all $k$ variables $X_{i_1}, ..., X_{i_k} \in B$ and their corresponding values in $b$. The probability of that branch is the probability of picking (uniformly at random) $X_{i_1}$ at the root node times the probability of testing, in that order, the remaining $X_{i_2}, ..., X_{i_k}$ variables in the sub-tree corresponding to the value $x_{i_1}$ of $X_{i_1}$ defined in $b$. Note that, by construction, it is certain that this particular sub-tree exists since the root node is split into $|\mathcal{X}_{i_1}|$ sub-trees. Then, the probability of testing $X_m$ at the end of this branch is the probability of picking $X_m$ among the remaining $p - k$ variables. By recursion, we thus have:

$$\frac{1}{p} \frac{1}{p-1} ... \frac{1}{p-k+1} \frac{1}{p-k} = \frac{(p-k)!}{p!} \frac{1}{p-k}$$

Since the order along which the variables appear in the branch is of no importance, $\alpha(B, b, X_m, p)$ actually includes all $k!$ equiprobable ways of building a branch composed of the variables and values in $B$ and $b$. Then, since a tree may at most contain a single such branch, whatever the order of the tests, the probabilities may be added up and it comes:

$$\alpha(B, b, X_m, p) = k! \frac{(p-k)!}{p!} \frac{1}{p-k} = \frac{1}{C_p^k} \frac{1}{p-k}$$

From the above expression, it appears that $\alpha(B, b, X_m, p)$ depends only on the size $k$ of $B$ and on the number $p$ of variables. As such, by grouping in the previous equation of $Imp(X_m)$ conditioning variable subsets $B$ according to their sizes and using the definition of conditional mutual information, $\alpha$ can be factored out, hence leading to the form foretold by Theorem 1:

$$Imp(X_m) = \sum_{k=0}^{p-1} \frac{1}{C_p^k} \frac{1}{p-k} \sum_{B \in \mathcal{P}_k(V^{-m})} I(X_m; Y|B).$$

$\square$

## C  Proof of Theorem 2

*Proof.* For any tree $T$, we have that the sum of all importances estimated by using an infinitely large sample $\mathcal{L}$ (or equivalently, by assuming perfect knowledge of the joint distribution $P(X_1,...,X_p,Y)$) is equal to $H(Y) - \sum_{l \in \ell(T)} p(l)H(Y|b(l))$, where $\ell(T)$ denotes the set of all leaves of $T$, and where $b(l)$ denotes the joint configuration of all input variables leading to leaf $l$. This is true because the impurities of all test nodes intervening in the computation of the variable importances, except the impurity $H(Y)$ at the root node of the tree, cancel each other when summing up the importances.

Since, when the tree is fully developed, $\sum_{l \in \ell(T)} p(l)H(Y|b(l))$ is obviously equal to the mean conditional entropy $H(Y|X_1,\ldots,X_p)$ of $Y$ given all input variables, this implies that for any fully developed tree we have that the sum of variable importances is equal to $I(X_1,\ldots,X_p;Y)$, and so this relation also holds when averaging over an infinite ensemble of totally randomized trees.  □

## D  Proof of Theorem 3

*Proof.* The proof directly results from the definition of irrelevance. If $X_i$ is irrelevant with respect to $V$, then $I(X_i;Y|B)$ is zero for all $B \subseteq V^{-i} \subset V$ and Equation 3 reduces to 0. Also, since $I(X_i;Y|B)$ is non-negative for any $B$, $Imp(X_i)$ is zero if and only if all its $I(X_i;Y|B)$ terms are zero. Since $Imp(X_i)$ includes all $I(X_i;Y|B)$ terms for $B \subseteq V^{-i}$, and since all of them are therefore null if $Imp(X_i) = 0$, $X_i$ is thus, by definition, irrelevant with respect to $V^{-i}$. $X_i$ is then also trivially irrelevant with respect to $V = V^{-i} \cup \{X_i\}$ since $I(X_i;Y|B \cup \{X_i\}) = 0$ for any $B$.  □

## E  Proof of Lemma 4

*Proof.* Let $X_i \notin V$ be an irrelevant variable with respect to $V$. For $X_m \in V$, $B \subseteq V^{-m}$, using the chain rules of mutual information, we have:

$$
\begin{aligned}
I(X_m,X_i;Y|B) &= I(X_m;Y|B) + I(X_i;Y|B \cup \{X_m\}) \\
&= I(X_i;Y|B) + I(X_m;Y|B \cup \{X_i\})
\end{aligned}
$$

If $X_i$ is irrelevant with respect to $V$, i.e., such that $I(X_i;Y|B) = 0$ for all $B \subseteq V$, then $I(X_i;Y|B \cup \{X_m\})$ and $I(X_i;Y|B)$ both equal 0, leading to

$$I(X_m;Y|B \cup \{X_i\}) = I(X_m;Y|B)$$

Then, from Theorem 1, the importance of $X_m$ as computed with an infinite ensemble of totally randomized trees built on $V \cup \{X_i\}$ can be simplified to:

$$
\begin{aligned}
Imp(X_m) &= \sum_{k=0}^{p-1+1} \frac{1}{C_{p+1}^k} \frac{1}{p+1-k} \sum_{B \in \mathcal{P}_k(V^{-m} \cup \{X_i\})} I(X_m;Y|B) \\
&= \sum_{k=0}^{p} \frac{1}{C_{p+1}^k} \frac{1}{p+1-k} \left[ \sum_{B \in \mathcal{P}_k(V^{-m})} I(X_m;Y|B) + \sum_{B \in \mathcal{P}_{k-1}(V^{-m})} I(X_m;Y|B \cup \{X_i\}) \right] \\
&= \sum_{k=0}^{p-1} \frac{1}{C_{p+1}^k} \frac{1}{p+1-k} \sum_{B \in \mathcal{P}_k(V^{-m})} I(X_m;Y|B) + \\
&\quad \hookrightarrow \sum_{k=1}^{p} \frac{1}{C_{p+1}^k} \frac{1}{p+1-k} \sum_{B \in \mathcal{P}_{k-1}(V^{-m})} I(X_m;Y|B) \\
&= \sum_{k=0}^{p-1} \frac{1}{C_{p+1}^k} \frac{1}{p+1-k} \sum_{B \in \mathcal{P}_k(V^{-m})} I(X_m;Y|B) +
\end{aligned}
$$

$$\hookrightarrow \sum_{k'=0}^{p-1} \frac{1}{C_{p+1}^{k'+1}} \frac{1}{p+1-k'-1} \sum_{B \in \mathcal{P}_{k'}(V^{-m})} I(X_m; Y|B)$$

$$= \sum_{k=0}^{p-1} \left[ \frac{1}{C_{p+1}^{k}} \frac{1}{p+1-k} + \frac{1}{C_{p+1}^{k+1}} \frac{1}{p-k} \right] \sum_{B \in \mathcal{P}_{k}(V^{-m})} I(X_m; Y|B)$$

$$= \sum_{k=0}^{p-1} \frac{1}{C_{p}^{k}} \frac{1}{p-k} \sum_{B \in \mathcal{P}_{k}(V^{-m})} I(X_m; Y|B)$$

The last line above exactly corresponds to the importance of $X_m$ as computed with an infinite ensemble of totally randomized trees built on $V$, which proves Lemma 4. $\square$

## F   Proof of Theorem 5

*Proof.* Let us assume that $V_R$ contains $r \le p$ relevant variables. If an infinite ensemble of totally randomized trees were to be built directly on those $r$ variables then, from Theorem 1, the importance of a relevant variable $X_m$ would be:

$$Imp(X_m) = \sum_{l=0}^{r-1} \frac{1}{C_r^l} \frac{1}{r-l} \sum_{B \in \mathcal{P}_l(V_R^{-m})} I(X_m; Y|B)$$

Let $X_i \in V \setminus V_R$ be one of the $p-r$ irrelevant variables in $V$ with respect to $V$. Since $X_i$ is also irrelevant with respect to $V_R$, using Lemma 4, the importance of $X_m$ when the ensemble is built on $V_R \cup \{X_i\}$ is the same as the one computed on $V_R$ only (i.e., as computed by the equation above). Using the same argument, adding a second irrelevant variable $X_{i'}$ with respect to $V$ – and therefore also with respect to $V_R \cup \{X_i\}$ – and building an ensemble of totally randomized trees on $V_R \cup \{X_i\} \cup \{X_{i'}\}$ will yield importances that are the same as those computed on $V_R \cup \{X_i\}$, which are themselves the same as those computed by an ensemble built on $V_R$. By induction, adding all $p-r$ irrelevant variables has therefore no effect on the importance of $X_m$, which means that:

$$Imp(X_m) = \sum_{k=0}^{p-1} \frac{1}{C_p^k} \frac{1}{p-k} \sum_{B \in \mathcal{P}_k(V^{-m})} I(X_m; Y|B)$$

$$= \sum_{l=0}^{r-1} \frac{1}{C_r^l} \frac{1}{r-l} \sum_{B \in \mathcal{P}_l(V_R^{-m})} I(X_m; Y|B)$$

$\square$

Intuitively, the independence with respect to irrelevant variables can be partly attributed to the fact that splitting at $t$ on some irrelevant variable $X_i$ should only dilute the local importance $p(t)\Delta i(t)$ of a relevant variable $X_m$ into the children $t_L$ and $t_R$, but not affect the total sum. For instance, if $X_m$ was to be used at $t$, then the local importance would be proportional to $p(t)$. By contrast, if $X_i$ was to be used at $t$ and $X_m$ at $t_L$ and $t_R$, then the sum of the local importances for $X_m$ would be proportional to $p(t_L) + p(t_R) = p(t)$, which does not change anything. Similarly, one can recursively invoke the same argument if $X_m$ was to be used deeper in $t_L$ or $t_R$.

## G   Proof of Proposition 6

*Proof.* The proof of Theorem 1 can be directly adapted to prove Proposition 6. If the recursive procedure is stopped at depth $q$, then it means that $B(t)$ may include up to $q-1$ variables, which is strictly equivalent to summing from $k = 0$ to $q-1$ in the outer sum of Equation 3. $\square$

# H  Proof of Proposition 7

*Proof.* Let us define a random subspace of size $q$ as a random subset $V_S \subseteq V$ such that $|V_S| = q$. By replacing $p$ with $q$ in Equation 3 (since each tree is built on $q$ variables) and adjusting by the probability

$$\frac{C_{p-k-1}^{q-k-1}}{C_p^q}$$

of having selected $X_m$ and the $k$ variables in the branch when drawing $V_S$ prior to the construction of the tree, it comes:

$$Imp(X_m) = \sum_{k=0}^{q-1} \frac{C_{p-k-1}^{q-k-1}}{C_p^q} \frac{1}{C_q^k} \frac{1}{q-k} \sum_{B \in \mathcal{P}_k(V^{-m})} I(X_m; Y|B)$$

The multiplicative factor in the outer sum can then be simplified as follows:

$$
\begin{aligned}
\frac{C_{p-k-1}^{q-k-1}}{C_p^q} \frac{1}{C_q^k} \frac{1}{q-k} &= \frac{\frac{(p-k-1)!}{(p-k)!(q-k-1)!}}{\frac{p!}{(p-q)!q!}} \frac{1}{C_q^k} \frac{1}{q-k} \\
&= \frac{(p-k-1)!q!}{(q-k-1)!p!} \frac{1}{C_q^k} \frac{1}{q-k} \\
&= \frac{q(q-1)...(q-k)}{p(p-1)...(p-k)} \frac{1}{C_q^k} \frac{1}{q-k} \\
&= \frac{q(q-1)...(q-k)}{p(p-1)...(p-k)} \frac{k!(q-k)!}{q!} \frac{1}{q-k} \\
&= \frac{1}{p(p-1)...(p-k)} \frac{k!(q-k)!}{(q-k-1)!} \frac{1}{q-k} \\
&= \frac{k!}{p(p-1)...(p-k)} \\
&= \frac{k!(p-k)!}{p!} \frac{1}{p-k} \\
&= \frac{1}{C_p^k} \frac{1}{p-k}
\end{aligned}
$$

which yields the same importance as in Proposition 6 and proves the proposition.  □

# I  Generalization to other impurity measures

In this appendix, we show that most of our results can be carried over to other impurity measures. We first sketch the proof that Theorems 1, 3 and 5 hold generically, and then discuss the case of some common impurity measures.

## I.1  Generalization of Theorems 1, 3 and 5

Let us consider a generic impurity measure $i(Y|t)$ and, by mimicking the notation used for conditional mutual information, let us denote by $G(Y; X_m|t)$ the impurity decrease for a split on $X_m$ at node $t$:

$$G(Y; X_m|t) = i(Y|t) - \sum_{x \in \mathcal{X}_m} p(t_x)i(Y|t_x),$$

where $t_x$ denotes the successor node of $t$ corresponding to value $x$ of $X_m$. The importance score associated to a variable $X_m$ (see Equation 2) is then rewritten:

$$Imp(X_m) = \frac{1}{N_T} \sum_T \sum_{t \in T : v(s_t) = X_m} p(t)G(Y; X_m|t).$$

As explained in the proof of Appendix B, conditioning over a node $t$ is equivalent to conditioning over an event of the form $B(t) = b(t)$, where $B(t)$ and $b(t)$ denote respectively the set of variables tested in the branch from the root to $t$ and their values in this branch. When the learning sample size $N$ grows to infinity, this yields the following population based impurity decrease at node $t$:

$$
\begin{aligned}
&G(Y; X_m | B(t) = b(t)) \\
=\ & i(Y | B(t) = b(t)) - \sum_{x \in \mathcal{X}_m} P(X_m = x | B(t) = b(t)) i(Y | B(t) = b(t), X_m = x).
\end{aligned}
$$

Again by analogy with conditional entropy and mutual information[2], let us define $i(Y|B)$ and $G(Y; X_m | B)$ for some subset of variables $B \subseteq V$ as follows:

$$
\begin{aligned}
i(Y|B) &= \sum_b P(B = b) i(Y | B = b) \\
G(Y; X_m | B) &= \sum_b P(B = b) G(Y; X_m | B = b) \\
&= i(Y|B) - i(Y|B, X_m)
\end{aligned}
$$

where the sums run over all possible combinations $b$ of values for the variables in $B$.

With these notations, the proof of Theorem 1 can be easily adapted to lead to the following generalization of Equation 3:

$$
Imp(X_m) = \sum_{k=0}^{p-1} \frac{1}{C_p^k} \frac{1}{p-k} \sum_{B \in \mathcal{P}_k(V^{-m})} G(Y; X_m | B).
$$

Note that this generalization is valid without any further specific constraints on the impurity measure $i(Y|t)$.

Let us now define as *irrelevant to $Y$ with respect to $V$* a variable $X_i$ for which, for all $B \subseteq V$, $G(Y; X_i | B) = 0$ (i.e. a variable that neither affects impurity whatever the conditioning). From this definition, one can deduce the following property of an irrelevant variable $X_i$ (for all $B \subseteq V$ and $X_m \in V$):

$$
G(Y; X_m | B \cup \{X_i\}) = G(Y; X_m | B).
$$

Indeed, by a simple application of previous definitions, we have:

$$
\begin{aligned}
&G(Y; X_m | B) - G(Y; X_m | B \cup \{X_i\}) \\
=\ & i(Y|B) - i(Y | B \cup \{X_m\}) - i(Y | B \cup \{X_i\}) + i(Y | B \cup \{X_i, X_m\}) \\
=\ & i(Y|B) - i(Y | B \cup \{X_i\}) - i(Y | B \cup \{X_m\}) + i(Y | B \cup \{X_i, X_m\}) \\
=\ & G(Y; X_i | B) - G(Y; X_i | B \cup \{X_m\}) \\
=\ & 0,
\end{aligned}
$$

where the last step is a consequence of the irrelevance of $X_i$.

Using this property, the proofs of Lemma 4 and Theorem 5 in Appendices E and F can be straightfor-wardly adapted, showing that, in the general case also, the MDI importance of a variable is invariant with respect to the removal or the addition of irrelevant variables.

Given the general definition of irrelevance, all irrelevant variables also get zero MDI importance but, without further constraints on the impurity measure $i$, there is no guarantee that all relevant variables (defined as all variables that are not irrelevant) will get a non zero importance. This property, and in consequence theorem 3, will be however satisfied as soon as the impurity measure is such that $G(Y; X_m | B) \geq 0$ for all $X_m \in V$ and for all $B \subseteq V$.

## I.2 Common impurity measures

Developments in the previous section show that all results in the paper remain valid for any impurity measure leading to non negative impurity decreases, provided that the definition of variable irrelevance is adapted to this impurity measure. The choice of a specific impurity measure should thus be guided by the meaning one wants to associate to irrelevance.

Measuring impurity with Shannon entropy, i.e., taking $i(Y|t) = H(Y|t)$ and $i(Y|B = b) = H(Y|B = b)$, one gets back all results in the paper. Given the properties of conditional mutual information, irrelevance for this impurity measure strictly coincides with conditional independence: a variable $X_i$ is irrelevant to $Y$ with respect to $V$ if and only if $X_i \perp Y|B$ for all $B \subseteq V$.

A common alternative to Shannon entropy for growing classification trees is Gini index (or entropy), which, in the finite and infinite sample cases, is written:

$$
\begin{aligned}
i(Y|t) &= -\sum_j p(j|t)(1 - p(j|t)) \\
i(Y|B = b) &= -\sum_j P(Y = j|B = b)(1 - P(Y = j|B = b)).
\end{aligned}
$$

Like Shannon entropy, this measure leads to non negative impurity decreases and the corresponding notion of irrelevance is also directly related to conditional independence.

The most common impurity measure for regression is variance, which, in the finite and infinite sample cases, is written:

$$
\begin{aligned}
i(Y|t) &= \frac{1}{N_t} \sum_{i \in t} (y_i - \frac{1}{N_t} \sum_{i \in t} y_i)^2 \\
i(Y|B = b) &= E_{Y|B=b}\{(Y - E_{Y|B=b}\{Y\})^2\}.
\end{aligned}
$$

Variance can only decrease as a consequence of a split and therefore, Theorem 3 is also valid for this impurity measure, meaning that only irrelevant variables will get a zero variance reduction. Note however that with this impurity measure, irrelevance is not directly related to conditional independence, as some variable $X_i$ can be irrelevant in the sense of our definition and still affects the distribution of output values.

## Footnotes

[1]To avoid problems, we suppose that all probabilities are strictly positive, without fundamental limitation.

[2]Note however that $G(Y; X_m | B)$ does not share all properties of conditional mutual information as for example $G(X_m; Y | B)$ might not be equal to $G(Y; X_m | B)$ or even be defined, depending on the impurity measure and the nature of the output $Y$.

## References

Cover, T. M. and Thomas, J. A. (2012). *Elements of information theory*. Wiley-interscience.