[Reviews · NeurIPS 2013]

Submitted by Assigned_Reviewer_3

This paper outlines theoretical methods for gauging the relevance of a particular variable when using forests of randomized trees (using p-ary splits and discrete variables). Given any (of the commonly used) impurity measures, several theorems are developed describing the MDI (importance) of a variable in an asymptotic case. The asymptotic condition is weakened and less rigorous (but still useful) claims are made about randomized trees that resemble those used in "typical" Random Forests. Finally, an experiment verifies some of the theorems and helps give some more intuition.

In terms of quality and clarity, the paper is very well written and is relatively easy to follow. In my opinion, some of the theorems could use a little more intuition and motivation.

I am not an expert in the field, but in terms of originality and significance, this seems to be the first paper attempting to analyze the importance of variables in the context of randomized trees - which is useful (however it would help to give more motivation for this point in the paper).

Since the theorems apply to asymptotic cases with fairly specific tree structures, it would greatly help to demonstrate (at least empirically) with real-world experiments how much the results deviate from the theory in the case of (smaller) finite trees, different number of splits per node, different variable subset sizes K, and different impurity measures. The 7-segment experiment is nice but doesn't help quantify the deviations.

I would have also liked to see a bit more motivation and description of prior work to help put this work in context.
Summary: A well written paper outlining several theoretical measures (albeit applicable in very specific conditions). Would help to have more motivation and some analysis of the deviation of the results from those using ideal conditions.

Submitted by Assigned_Reviewer_5

This paper attempts to analyze theoretically a popular measure of variable importance, computed by random forests (RF) or extra-trees. The gap between the practical success of RF, in particular for feature selection, and its theoretical understanding is so huge that the author's efforts should be praised.

The main results of this paper show that, in a particular RF model (discrete attributes, totally randomized and fully developed), we can obtain an exact estimate of the RF importance in terms of mutual information. This is a very nice result, which allows to understand what is estimated by the RF algorithm. The extension to slightly more realistic models (pruned RF, or RF where at each node, the best among K candidate splits is chosen) shows that the situation is more complicated in these cases, but give useful hints at what are the mechanisms which play a role. The experiments illustrate nicely the main findings of the paper.

With a little more effort, one assumes that results (bounds) for a finite mixture of trees could be obtained too.

There is still a gap between the theory of RF and their practical use, but this paper offers a nice improvement over the state-of-the-art which is almost void for RF feature importance. It would certainly be of interest for people interested in understanding the good practical behavior of RF.
Summary: A very nice theoretical analysis of the variable importance computed by some simplified models of RF. Only caveat is that extending the theory to more realistic models of RF is difficult, but this paper is among the first to say something about it.

Submitted by Assigned_Reviewer_6

Variable importance measures are often used to highlight key predictor variables in tree-based ensemble models. However, there has generally been a lack of a theoretical understanding of these measures. In this paper, the authors study the theoretical properties of the Mean Decrease Impurity (MDI) importance measures (such as the Gini importance). Through most of the paper they use the Shanon entropy as the impurity measure but show that the theorems and results are applicable to any impurity measure. They begin with an asymptotic analysis of totally randomized fully-developed tree ensembles learned using an infinitely large ensemble. They show that under these conditions the variable importance for all variables provide a three-level decomposition of the information contained in the set of input variables about the output variable. First, they show that the information decomposes as a sum of importance scores of all input variables. Second, the importance of each variable decomposes as a sum over all degrees of interactions of that variable with other variables. The third level of decomposition for each degree is over all possible combinatorial interactions of that degree. Next, the authors show that that the MDI importance of a variable is equal to zero if and only if the variable is irrelevant and that the MDI importance of a relevant variable is invariant with respect to the removal or the addition of irrelevant variables. The authors then analyze the properties of the variable importance measures in depth-pruned ensembles of randomized trees. They show that as long as the pruning depth is greater than the total number of true relevant variables, these relevant variables still obtain a strictly positive importance score (albeit different from their values in fully-developed trees) and the irrelevant variables continue to obtain an importance of zero. Finally, for the case of ensembles consisting of non-totally randomized trees such as random forests, where top scoring predictors are selected at splits from a random subset of predictors, they show strong masking effects by the strongest variables resulting in over or under-estimation of importance scores. They finally show the relevance of their results through a simple yet insightful example of digit recognition.

The paper is very accessible and written in a very understandable manner. At the same time, the theorems and results discussed are very interesting and provide some nice theoretical insights into variable importance measures from idealized asymptotic models to more realistic models. The masking effects by the strongest predictors and the inability of the variable importance to appropriately account for interactions and dependencies between variables for random-forest like models are very relevant to practical analyses and may be amplified even more in cases involving correlated or partially correlated variables. I think the paper lays down some nice initial results and a general framework to analyze more realistic models in the future and potentially point to better strategies for learning trees or computing more accurate importance scores. The paper will definitely be of interest to the general machine learning community.
Summary: The paper provides an insightful theoretical analysis of variable importance measures in tree-based ensemble models ranging from idealized asymptotic models to realistic random-forest like models. It provides a key framework to study these types of measures and potentially improve them.
Author Feedback

Author rebuttal: Dear reviewers,

We would like to thank you for your positive reviews. We agree with the assessment of our submission.

Best regards,

The authors.